# Augmentative Plating versus Exchange Intramedullary Nailing for the Treatment of Aseptic Non-Unions of the Femoral Shaft—A Biomechanical Study in a Sawbone^TM^ Model

**DOI:** 10.3390/jpm13040650

**Published:** 2023-04-10

**Authors:** Matthias Georg Walcher, Robert E. Day, Markus Gesslein, Hermann Josef Bail, Markus S. Kuster

**Affiliations:** 1OC Würzburg, Oeggstr. 3, 97070 Würzburg, Germany; 2Department of Orthopedics and Traumatology, Paracelsus Medical University, Breslauer Straße 201, 90471 Nuremberg, Germany; 3Health Technology Management Unit, Royal Perth Hospital, University of Western Australia, 197 Wellington Street, Perth 6000, Australia; 4Perth Orthopaedic Sports Medicine Centre, The University of Western Australia, 31 Outram Street, Perth 6005, Australia

**Keywords:** non-union long bone, biomechanics, additional plate, nail in situ, non-union revision

## Abstract

Background: Non-unions after intramedullary nailing of femoral shaft fractures are a significant problem. Treatment options such as augmenting with plates or exchange nailing have been proposed. The ideal treatment remains controversial. Methods: Augmentative plating using a 4.5 mm LCP or a 3.2 mm LCP leaving the nail in situ was tested biomechanically and compared to exchange intramedullary nailing in a Sawbone^TM^ model of a femoral shaft non-union. Results: The difference of fracture gap motion in axial testing was small. In rotational testing, the exchange nail allowed for the largest amount of motion. The 4.5 mm augmentative plate was the most stable construct in all loading conditions. Conclusions: Augmentative plating using a 4.5 mm LCP plate while leaving the nail in situ is biomechanically superior to exchange intramedullary nailing. A small fragment 3.2 mm LCP is undersized and does not reduce fracture motion sufficiently in a femoral shaft non-union.

## 1. Article Summary

Article focus:The main revision options for aseptic non-unions in femoral shaft fractures after intramedullary nailing without a bony defect are revision intramedullary nailing and augmentative plating.Revision intramedullary nailing and augmentative plating using plates of different sizes were evaluated biomechanically in a Sawbone^TM^ model of an aseptic non-union of the femoral shaft.

Key messages:Augmentative plating using a 4.5 mm LCP leaving the nail in situ is more stable than exchange intramedullary nailing.Using a 3.2 mm LCP does not reduce fracture motion sufficiently.

Strengths and limitations of this study:Strengths: biomechanically based treatment recommendations for the treatment of aseptic femoral shaft non-unions after intramedullary nailing are provided.Limitations: no cadaver model was used, but a sawbone model was used. This is less realistic than cadaver bones but has the advantage of uniformity. Only two different plates were used for evaluation.

## 2. Introduction

Non-union after intramedullary nailing of fractures of diaphyseal long bones is a significant complication, leading to a long period of incapacity, implant failure, revision surgery and, in the worst case, even to the loss of a limb [1]. It is crucial to manage non-unions effectively to ensure the best chances of bone healing [1,2], especially as there is a relevant percentage of femoral non-unions which does not heal without more additional procedures, or does not heal at all [3]. The traditional approach to non-unions of long bones after intramedullary nailing without osseous defect or infection is a removal of the nail, reaming of the medullary canal, and stabilization, typically with a larger diameter nail [1,2,4]. Recently, the concept of leaving the nail in situ and adding stability with an additional plate fixation has gained popularity with promising results in the literature [5,6,7,8,9,10,11,12,13,14]. The concept behind this is that, for hypertrophic non-unions, the stability of fracture fixation is insufficient. Therefore, adding extra stability and ideally compression of the non-union is necessary to achieve bony union. Exchange nailing still seems to be the most popular amongst these treatment options. The idea of augmenting an osteosynthetic construct and adding extra stability by doing so has also been described in other fields of orthopedic trauma, like trochanteric femur fractures, where cement augmentation has been found to be safe and effective in osteoporotic fractures [15]. It needs to be stated that a lack of mechanical stability is not the only reason for a non-union. In femur fractures, the same reasons, like in any other fracture, can cause a non-union, namely too much motion at the fracture site, avascularity, bony gapping, and infection [16]. This explains the different shapes of non-unions—hypertrophic, atrophic, and oligotrophic. Bony gapping needs to be addressed with a proper operative technique, potentially in combination with a bone graft. Infectious non-unions typically need to be treated with thorough surgical debridement, rigid fixation often using external fixators, and a tailored antibiotic treatment. Avascularity is a risk factor for the non-union of femoral fractures. However, it is important to note that the effect of anatomic reduction and stability of fracture fixation can often outweigh this, allowing for a bony healing [16]. Therefore, proper osteosynthesis is in the center of the treatment of non-unions after fractures of every origin—there is basically no fracture healing without stability. To the best of our knowledge, no biomechanical investigation has been published about the effects of an augmentative plate. The aim of the present study was to evaluate the biomechanical rationale behind the concept of leaving the nail in situ while adding an additional lateral plate stabilization. The stability of additional plates of different sizes leaving the original nail in situ was compared to an exchange intramedullary nail with a larger diameter. Clinically, there are indications that the combination of an intramedullary nail and an augmentative plate offers excellent stability. The group around Passias recommends using a nail in combination with a lateral locking plate in distal femur fractures in patients that are at a higher risk for non-union or who require extra robust fixation in primary fracture care [17].

## 3. Materials and Methods

Fifteen large fourth generation composite femur models (Sawbones^TM^, Pacific Research Labs, Vashon, WA, USA) with a diaphyseal outer diameter of 32 mm were used. Composite femora were used, as they have greater uniformity than cadaver bone and have been thoroughly evaluated as bone analogues [18]. To approximate a model of an aseptic non-union of the femoral shaft with the nail loose in the medullary canal, the composite femora were reamed to an inner diameter of 16 mm. A transverse fracture was created 130 mm proximal to the distal femoral end in the transition zone between the distal and the middle third of the femur with a saw. The fractures were reduced and fixed with an intramedullary nail. Three groups were tested: An exchange nail with 14 mm diameter (big nail), then a 3.2 mm augmentative plate (3.2 plate) and a 4.5 mm augmentative plate (4.5 plate), both with the 11 mm nail in situ. Five femora were used in each group.

In the exchange intramedullary nail group, five femora were fixed with a 14 mm femoral nail (AFN, Synthes^TM^, Westchester, PA, USA) (Figure 1A). It is a hollow femoral nail design. Four regular locking bolts were used. Distally one dynamic and one static locking bolt were inserted, and proximally the regular locking option was chosen with two locking bolts from lateral to medial. The 14 mm diameter nail had a tight fit in the medullary canal, which recreated the clinical situation of a tight exchange nail. 

In the augmentative plate group, an 11 mm femoral nail (AFN, Synthes^TM^, Westchester, PA, USA) was inserted to represent a non-union with an initially tight nail that had become loose. No distal locking bolts were inserted to simulate removing the distal locking bolts to dynamize the nonunion, which is typically part of the augmentative plating procedure. Following that, a 9-hole 3.2 stainless steel LCP plate (Synthes^TM^, Westchester, PA, USA) was applied to five femora on the lateral side, and a 7-hole 4.5 steel LCP plate (Synthes^TM^, Westchester, PA, USA) was applied to another five femora, again on the lateral side, to match the clinical situation of an augmentative plate, inserted from the lateral side. Both groups used three non-angular stable screws on either side of the fracture (Figure 1B). The screws were drilled and inserted eccentrically to create compression at the fracture site. The screws were inserted as non-angular-stable. The dynamic compression holes of the plate were used in a typical AO-way, not only to add stabilization, but to achieve compression at the fracture site. Ideally, the screws would be positioned in different directions—some anterior and some posterior of the nail—to position the material “around the nail”. Aiming to recreate a model as close to the clinical situation as possible, we inserted the screws where they could be positioned safely bicortically, be it anterior or posterior of the nail. This resembles the situation in operation theatre, where there is typically only a very narrow corridor for screw and plate placement around the nail, not allowing readily to decide to position the screws anterior or posterior to the nail, in particular in atrophic non-unions with little vital bone available.

A total of 15 artificial composite femora, one 3.2 LCP, one 4.5 LCP, one 11 mm, and one 14 mm AFN, were used for testing.

For image correlation strain measurement, the anterior surface of the bone was sprayed with a speckle pattern (Figure 1C). The system allowed full-field 3D-surface strain and motion detection down to 200 µstrain in this set-up (VIC-3D™, Limess, Correlated Solutions) (Figure 1D). The construct was mounted to a servohydraulic biaxial testing machine (Instron 8874) with a 25 kN/100 Nm load cell. The Sawbone femora were potted in PMMA distally and held by a universal joint that constrained translation and axial torsion but allowed rotation in the other two axes. Proximally, the femoral head was held in a custom fixture that was rigidly attached to the load cell. The axis of the bone was aligned in 7° valgus to the rotational axis of the testing machine to recreate a loading situation that is as close to physiological loading as possible [19]. Of course, the model remained crude, as the forces of the surrounding muscles and soft tissue were not included (Figure 1E). 

Axial loading and torsional loading were applied for 26 cycles and the mean peak strain was calculated on the bone surface in the region of the artificial fracture on the anterior surface of the femur and averaged over 20 cycles. The first three and the last three cycles were not included in the calculations to avoid the starting and stopping ramps of the test. The regions on the anterior femur immediately above and below the fracture were analyzed, as this would allow for the best assessment of the stability of the fracture fixation. 

The testing started with a ramp to 10 N of axial compression over 3 s. Following that, 26 sinusoidal cycles of axial compression from 10 N to 870 N at 0.25 Hz were applied. This was followed by a ramp over 3 s to 20 N axial preload and 0 Nm torsion to ensure consistent starting conditions. Sinusoidal cyclic rotation was then applied, ranging from −6 Nm to 6 Nm at 0.25 Hz for 26 cycles. Again, for analysis, the first three and the last three cycles were discarded as the system took time to come to a full load and to a stop. The number of cycles was determined from pilot studies to give stable average measurements. The orientation and neck offset were chosen to match the study of Bergmann [19]. The loading patterns were chosen according to the study of Schneider et al., who calculated the forces acting in a femur during healing after fracture [20]. They found forces in the axial direction up to 300 N in 250 N partial weightbearing, and up to 900 N in single leg stance. The torsional moment they described was up to 6 Nm in 250 N partial weightbearing, and up to 10 Nm during torsion testing [20]. We aimed to recreate the lower range of physiological loading conditions. The constructs were compared separately for each loading condition using one-way ANOVA. The Q-Q plot was checked before testing to confirm that the data were normally distributed. The construct had a significant effect on both the axial and torsion loading results, so Tukey’s honestly significant difference test was applied to look for the differences between the constructs. The level of significance was set at 5% for all tests. All analyses were performed using R version 3.1.2 running on OSX 10.9.5.

## 4. Results

Relative motion between the two fragments in three different directions were measured to judge the rigidity of the constructs. Figure 2 and Table 1 display the results. Us is fracture motion in the proximal/distal direction, Vs is in the medial/lateral direction, and Ws is in the anterior/posterior direction. The total interfragmentary motion D is the vector sum of the motions in the different directions. 

### 4.1. Axial Loading

Analyzing the total fracture gap motion, the large additional plate was the most rigid, followed by the revision intramedullary nail and the small plate, which had the largest motion. The differences between the treatment groups were all significant (see Table 2).

In the compression/distraction direction (Us), both the group with the revision intramedullary nail (big nail) and the large plate (4.5 plate) had much smaller motion at the fracture gap than the small plate (3.2 plate). The axial motion of the small plate group was significantly higher (*p* < 0.05) than either of the other groups. In the medial/lateral direction (Vs), the large additional plate group showed significantly less fracture motion (*p* < 0.05) than the other constructs, which both displayed comparable amounts of movement. In the anterior/posterior direction (Ws) all treatment groups had comparable small motion, without significant differences.

### 4.2. Torsional Loading

Comparing the sums of movement in the fracture gap, the large plate group had the smallest motion, followed by the small plate group, while the exchange nail was the least rigid construct in torsion. The differences between the treatment groups were all statistically significant. Table 3 displays the differences between the constructs in torsion. In the proximal/distal direction (Us), all three constructs showed a small amplitude of motion. The large plate was significantly more stable (*p* < 0.05) than the two other constructs, which both had a similar range of fracture motion. The large plate group was also the most robust construct in the medial/lateral direction (Vs), followed by the small plate group and then the revision intramedullary rod. The differences between the treatment groups were statistically significant (*p*< 0.05). In the anterior/posterior direction (Ws) the large plate proved to be the most stable construct with the smallest fracture motion, followed by the revision intramedullary nail and the small plate. The differences between the treatment groups were all statistically significant.

Figure 3 shows the overall displacement by construct for both axial and torsion loading. In summary, all three osteosynthetic constructs showed sufficient stability under axial loading conditions with little motion in the fracture gap. Motion during torsional loading was larger than in axial compression for all constructs. In compression the difference between the constructs was generally small, with the exchange nail being superior to the 3.2 mm plate by 0.2 mm and marginally worse than the 4.5 mm plate (about 0.1 mm). A much larger motion was found in torsional loading, with the exchange nail having the largest motion. This was nearly 2 mm more than the 4.5 mm plate and over 1 mm more than the 3.2 mm plate. All of the differences in motion between the different constructs were statistically significant. The 4.5 mm plate construct had the least interfragmentary motion in all cases.

## 5. Discussion

There are two main options in orthopedic literature for the treatment of aseptic femoral non-unions without bony defect after intramedullary nailing which are discussed in the orthopedic literature: augmentative plating or exchange nailing. Exchange nailing is usually combined with reaming of the medullary canal and conversion to a larger diameter intramedullary nail [2]. 

Brinker summarized the results of the current literature for femoral non-unions after intramedullary nailing [2]. He identified several papers on augmentative plating whilst leaving the nail in situ [5,6,7,10,11,12,13,14]. These publications reported the results of a total of 147 cases, with a healing rate of 100%. The group around Zhang could confirm these excellent results with a 100% healing rate in 39 patients with a non-infectious non-union after intramedullary nail fixation for long bone fractures in the lower extremity that were treated with augmentative plating [21]. The authors pointed out the easy operative technique with a short operation time. The majority of the authors used a 4.5 mm DCP, as was done in the present biomechanical testing. 

Uliana and coworkers performed a multicenter study, including 22 cases treated with plate augmentation for a femoral shaft aseptic non-union of a femur fracture with an intramedullary nail retained in situ [22]. These patients were followed for an average of 23.5 months. Among these 22 patients were 10 patients with a vascular non-union. Nonetheless, bony healing could be achieved in 19 patients, and all patients were doing well clinically. Just recently, Perisano published a literature review looking into plate augmentation in aseptic femoral shaft non-union after intramedullary nailing [23]. He included 24 studies with a total of 502 patients. Overall, 200 had an atrophic non-union, 123 had a hypertrophic non-union, and in 179 patients the type of non-union was not mentioned. The plate used the most was a DCP (dynamic compression plate). In 98% of the cases, osseous union could be achieved. Functional recovery was good, and the rate of complications was low [23].

There were eight publications investigating exchange intramedullary nailing for aseptic femoral shaft non-unions [2]. A total of 266 cases were reported with a healing rate of 89% [10,24,25,26,27,28,29,30].

Most of these authors did over-ream the medullary canal by 1–2 mm in the revision procedure and used a revision nail that was at least 1 mm wider than the original nail. Of particular interest is the work of Yang et al. [29]. They evaluated isthmic and non-isthmic non-unions separately. In the isthmic non-unions, 87% healed with exchange intramedullary nailing, while in the non-isthmic non-unions only 50% healed. This is in agreement with our biomechanical results—that rotational stability is an important issue—which is more prevalent in the wide, non-isthmic portion of the femur. Lai and his group looked retrospectively into 96 patients undergoing either augmentative plating or exchanging reamed nailing [31]. The overall union rate was 70.8%, even with the exclusion of hypertrophic non-unions. The union rate was significantly higher in the augmentative plate group, and the operating time was significantly shorter. Looking into the location of the non-union, the union rate was comparable for isthmic non-unions, but was significantly smaller for the augmentative plates in non-isthmic non-unions. These results were confirmed by Ru, who looked into the results of 190 non-unions of the femur after intramedullary nailing. They were treated either with exchange reamed intramedullary nailing or augmentative plating [32]. Augmentative plating could obtain a higher union rate and a shorter time to union. The intraoperative blood loss, like the complication rate, was higher for the exchange nail group. 

Nonetheless, Kim and Coworkers could achieve a 100% healing rate in their patient collective of infraisthmal femoral non-unions [4]. They treated 18 patients with a non-union of a femur shaft fracture in the infraisthmic portion, not only with an exchange intramedullary nail, but enhanced the stability of their construct by adding a Poller screw and additional interlocking screws. These results confirm that exchange nailing on its own is just biomechanically not stable enough in the non-isthmic portion of the femur. 

Park and his colleagues treated 17 patients with a hypertrophic non-union of the long bones with an augmentative locking compression plate [33]. They did no fracture site exposure, implant removal, or bone grafting. Still, all cases achieved osseous healing, proving the mechanical stability of augmentative plating. Sancheti combined exchange nailing with augmentative plating in 70 patients with femoral diaphyseal non-unions to profit from the benefits of both treatment options. Overall, 46 cases had a hypertrophic non-union and 24 had an atrophic non-union. All patients received autologous bone grafting and raising of periosteal flaps. The healing rate was 100% at an average time of 16.77 weeks. 

It is of particular interest that the concept of augmentative plating also worked in atrophic nun-unions. In atrophic non-unions, a disturbed vascularity of the fracture regions must be assumed. The endosteal blood supply is already disturbed by the initial nailing, so plating could be potentially detrimental by inhibiting the periosteal blood supply. Obviously, the extra stability outweighs the potential side effects on vascularity [16]. These positive effects of augmentative plating in atrophic non-unions could also be observed in other studies [22,23,31].

Kontakis proposed using a primary nail/augmentative plate construct, even in primary fracture cases, and in fractures of the distal femur in geriatric, obese, or osteoporotic patients, to allow for immediate weight-bearing and range of motion exercises after fracture fixation [34].

In summary, both augmentative plating leaving the nail in situ and exchange intramedullary nailing are valid options to treat aseptic femoral non-unions. The sparse literature shows a higher success rate for augmentative plating in aseptic non-unions of the femoral shaft after intramedullary nailing, especially in the non-isthmic portion of the femur. These findings are confirmed by several recent meta-analyses comparing these two treatment options, which describe a higher rate of union, a shorter healing time, less intraoperative blood loss, and fewer complications for augmentative plating [35,36,37]. Ru could establish that revision with an exchange, reamed intramedullary nail instead of augmentative plating is a risk factor for the development of another non-union [38]. Our results support these findings by supplying a biomechanical explanation. In biomechanical testing conditions, augmentative plating, in particular with a 4.5 mm LCP, is clearly more stable than exchange nailing in every tested loading condition. In the axial compression testing the differences between the different constructs were small, while torsional loading produced larger motion in the fracture gap. While the exchange intramedullary nail was stronger than the small augmentative plate in axial compression, the exchange intramedullary nail was clearly the least stable in rotational testing. In summary, using a 4.5 mm LCP as an augmentative plate leaving the nail in situ in aseptic femoral shaft non-unions is biomechanically superior in both torsion and axial compression compared to an exchange nail construct. Using a 3.2 mm LCP gives larger fracture motions and seems to be an undersized construct. 

## 6. Strengths

The study presents biomechanically based recommendations for the treatment of aseptic femoral non-unions after intramedullary nail stabilization.

These results can help orthopedic surgeons to reduce postoperative complications and reach an ideal rate of unions after revision surgery. 

## 7. Limitations

We did not use cadaveric femora but used a sawbone model. This is a less realistic model, but the uniformity of the specimens allows for comparison between constructs without introducing the large variability of cadaveric bone qualities. We only used a 3.2 LCP and a 4.5 LCP. No other plates of different specifications and lengths were used for comparison, and no angular stable locking plates were used.

## 8. Conclusions

In aseptic femoral shaft non-unions, a 4.5 mm LCP as an augmentative plate leaving the nail in situ is biomechanically superior in terms of stability in both torsion and axial compression to an exchange nail construct using a bigger nail. A 3.2 mm LCP seems to produce an undersized construct.

## Figures and Tables

**Figure 1 jpm-13-00650-f001:**
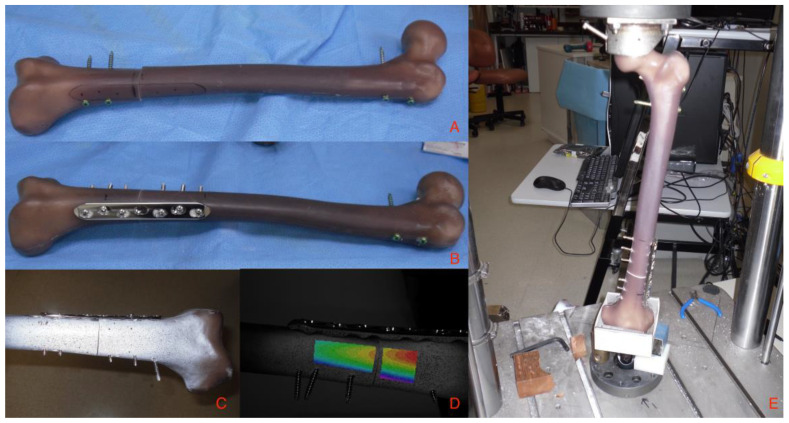
Test setting. (**A**) Preparation of the sawbone femora, with the nail in situ and an artificial fracture applied. (**B**) An additional plate is mounted. (**C**) The anterior surface of the bone is sprayed with a speckle pattern. (**D**) The system allows full-field 3D-surface strain and motion detection down to 200 µstrain. (**E**) Sawbone femora are potted in PMMA distally and held by a universal joint that constrained translation and axial torsion but allow rotation in the other two axes. The femoral head is held in a custom fixture rigidly attached to the load cell. The axis of the bone is aligned in 7° valgus to the rotational axis of the testing machine to recreate a loading situation as close to physiology as possible.

**Figure 2 jpm-13-00650-f002:**
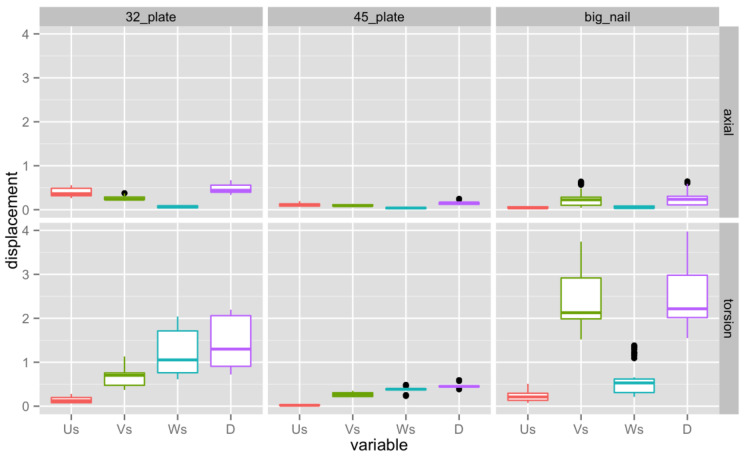
Us is fracture motion in the proximal/distal direction, Vs is in the medial/lateral direction, and Ws is in the anterior/posterior direction. The total interfragmentary motion D is the vector sum of the motions in the different directions. The 3.2 plate stands for the small additional plate, the 4.5 plate stands for the larger additional plate, and the big nail stands for the exchange nail. The different colors have been added to allow for a clearer determination of the information. In every column the middle horizontal line is the median. The box goes from the 25^th^ percentile to the 75^th^ percentile and the black dots mark the extreme values.

**Figure 3 jpm-13-00650-f003:**
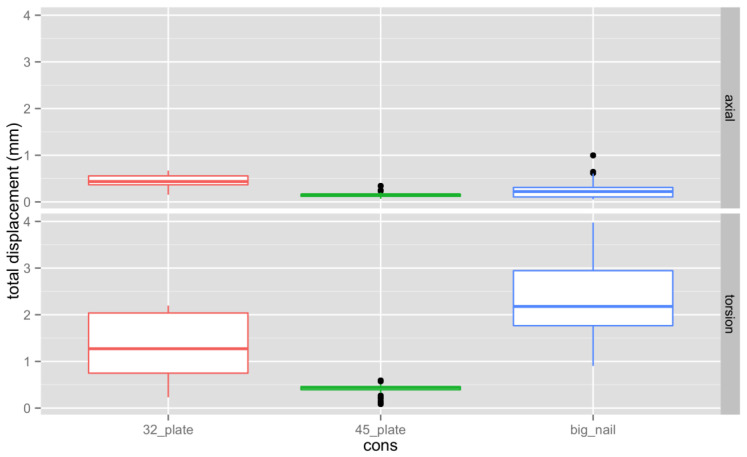
Overall displacement by construct for both axial and torsion loading. The different colors have been added to allow for a clearer determination of the information. In every column the middle horizontal line is the median. The box goes from the 25th percentile to the 75th percentile and the black dots mark the extreme values.

**Table 1 jpm-13-00650-t001:** Mean total interfragmentary motion in mm by construct and loading directions. The big nail stands for exchange medullary nail, the 3.2 plate stands for small augmentative plate, and the 4.5 plate stands for large augmentative plate.

Construct	Axial Loading	Torsional Loading
3.2 plate	0.46	1.33
4.5 plate	0.16	0.43
big nail	0.26	2.42

**Table 2 jpm-13-00650-t002:** Differences between constructs in axial loading. The big nail stands for exchange medullary nail, the 3.2 plate stands for small additional plates, and the 4.5 plate stands for large additional plates.

	Difference	99.9% CI	*p* adj
big nail versus 4.5 plate	0.0992	0.0847–0.114	<0.01
3.2 plate versus big nail	0.2063	0.1917–0.221	<0.01
3.2 plate versus 4.5 plate	0.3055	0.2910–0.320	<0.01

**Table 3 jpm-13-00650-t003:** Differences between constructs in torsion. The big nail stands for the exchange medullary nail, the 3.2 plate stands for the small additional plate, and the 4.5 plate stands for the large additional plate.

	Difference	99.9% CI	*p* adj
big nail versus4.5 plate	0.891	0.825–0.957	<0.01
3.2 plate versus big nail	1.094	1.028–1.160	<0.01
3.2 plate versus 4.5 plate	1.985	1.919–2.051	<0.01

## Data Availability

The data is available at the Health Technology Management Unit, Royal Perth Hospital, University of Western Australia.

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
