# Peer review of "Augmentative Plating versus Exchange Intramedullary Nailing for the Treatment of Aseptic Non-Unions of the Femoral Shaft—A Biomechanical Study in a SawboneTM Model"

_jpm, 2023, doi:10.3390/jpm13040650_

Round 1
Reviewer 1 Report
This study aimed to evaluate the biomechanical rationale behind the concept of leaving the nail in situ while adding an additional lateral plate stabilization for non-unions after intramedullary nailing of femoral shaft fractures. This is a very interesting study, but there are some problems that should be further addressed.
1. In the method section, please describe the method of screw insertion of additional steel plate sets in detail. It is recommended to use an illustration, which is more conducive to readers' understanding. As mentioned in the article, screws were inserted eccentrically. We imagine that, multiple screws located on one side of the intramedullary nail and the screws located in different directions of the intramedullary nail (such as one is located in front of the nail, and another is located behind the nail), and the results of the two may be different.
2. In the augmentative plate group, 9-hole 3.2 stainless steel LCP plate and 7-hole 4.5 steel LCP plate were used for biomechanical testing. Other plates of different specifications and lengths are not used for comparison, which may also be one of the limitations of this study.
3. It is recommended to add references for the last three years.
Author Response
Reviewer 1
This study aimed to evaluate the biomechanical rationale behind the concept of leaving the nail in situ while adding an additional lateral plate stabilization for non-unions after intramedullary nailing of femoral shaft fractures. This is a very interesting study, but there are some problems that should be further addressed.
- In the method section, please describe the method of screw insertion of additional steel plate sets in detail. It is recommended to use an illustration, which is more conducive to readers' understanding. As mentioned in the article, screws were inserted eccentrically. We imagine that, multiple screws located on one side of the intramedullary nail and the screws located in different directions of the intramedullary nail (such as one is located in front of the nail, and another is located behind the nail), and the results of the two may be different.
Thank you for pointing that out. We have clarified that and described it in detail in line 152ff. We would guess that the orthopedic readers are familiar with the concept of a dynamic compression plate, so that we were confident to get along without another illustration.
- In the augmentative plate group, 9-hole 3.2 stainless steel LCP plate and 7-hole 4.5 steel LCP plate were used for biomechanical testing. Other plates of different specifications and lengths are not used for comparison, which may also be one of the limitations of this study.
Thanks for this comment. Of course, you are right, and I included this in our limitations.
- It is recommended to add references for the last three years.
Thanks for pointing out, we have given our literature a thorough work-over.
Reviewer 2 Report
A straightforward study which aims to reproduce the clinical situation experienced by surgeons, in the eyes of engineers. The discussion is not nuanced enough from a surgical perspective with regards the role of biology and motion in non-union ie this is more relevant in hypertrophic non-union where the aim is to stabilise or reduce relative movement. This compares to atrophic or delayed union, where biological factors may be more relevant and this may have particular relevance when discussions around reaming are concerned. This discussion could be broadened to include the fact that plating may actually be further detrimental to the periosteal blood supply, which may be more important if the endosteal blood supply has been traumatised by the nailing.
Other minor points:
1. Was the nail solid or hollow
2. their model does not 'recreate physiologic loading'. This needs rephrasing to state that it is crude and non-physiological but simpler than including all the muscle forces and used by others etc.
3. It looks untidy to have different group names eg Big nail / EN / exchange intramedullary nail. Uniformity such as - 32 Plate / 45 Plate / Exchg Nail - might be clearer.
4. References: It might be helpful to have a reference to support the frequency of non union, such as 'Plate Augmentation in Aseptic Femoral Shaft Nonunion after Intramedullary Nailing: A Literature Review' by Carlo Persano. Maybe a reference by Egol KA might be appropriate too, such as their view on the difficulty of subtroch femoral fractures as they seem to be on the current lecture circuit.
Author Response
Reviewer 2
Comments and Suggestions for Authors
A straightforward study which aims to reproduce the clinical situation experienced by surgeons, in the eyes of engineers. The discussion is not nuanced enough from a surgical perspective with regards the role of biology and motion in non-union ie this is more relevant in hypertrophic non-union where the aim is to stabilise or reduce relative movement. This compares to atrophic or delayed union, where biological factors may be more relevant and this may have particular relevance when discussions around reaming are concerned. This discussion could be broadened to include the fact that plating may actually be further detrimental to the periosteal blood supply, which may be more important if the endosteal blood supply has been traumatised by the nailing.
That is a very valid point, thanks for putting our eyes on this. We have included it in the discussion in the lines 390 ff.
Other minor points:
- Was the nail solid or hollow
Thanks for asking this important question. We have added the information in the material and methods section of the paper.
- their model does not 'recreate physiologic loading'. This needs rephrasing to state that it is crude and non-physiological but simpler than including all the muscle forces and used by others etc.
That is of course right. We have made that clear in line 179 ff.
- It looks untidy to have different group names eg Big nail / EN / exchange intramedullary nail. Uniformity such as - 32 Plate / 45 Plate / Exchg Nail - might be clearer.
Thank you for pointing that out. We have changed the paper accordingly.
- References: It might be helpful to have a reference to support the frequency of non union, such as 'Plate Augmentation in Aseptic Femoral Shaft Nonunion after Intramedullary Nailing: A Literature Review' by Carlo Persano. Maybe a reference by Egol KA might be appropriate too, such as their view on the difficulty of subtroch femoral fractures as they seem to be on the current lecture circuit.
Thanks for these suggestions. Indeed, both recommendations proved to be very helpful. We have included the paper of Perisano, and a reference of the group of Kenneth Egol (deRogatis et al). As our model does not represent subtrochanteric fractures, we did concentrate on femoral shaft non-unions in our discussion.
Reviewer 3 Report
Dear Editor
Thank you for giving me the opportunity to review the manuscript " Augmentative plating versus exchange intramedullary nailing for the treatment of aseptic non-unions of the femoral shaft – a biomechanical study in a sawboneTM model"
 The topic discussed is clinically relevant.
General
A revision of the English language by a native speaker must be performed. PLEASE DO CORRECT SPELLING MISTAKES
Introduction
Well written, Update the literature such as Stramazzo et al.(PMID: 33717919), Napoli et al. (PMID: 18977467), Ciolli et al. (PMID: 33480224)
Methods
Well written. Specify what type surgery lower limb.
Discussion
well written
Author Response
Reviewer 3
Dear Editor
Thank you for giving me the opportunity to review the manuscript " Augmentative plating versus exchange intramedullary nailing for the treatment of aseptic non-unions of the femoral shaft – a biomechanical study in a sawboneTM model"
 The topic discussed is clinically relevant.
General
A revision of the English language by a native speaker must be performed. PLEASE DO CORRECT SPELLING MISTAKES
Thanks for pointing this out. One of our authors is a native speaker and corrected the paper prior to submission. Nonetheless we gave the paper another language check.
Introduction
Well written, Update the literature such as Stramazzo et al.(PMID: 33717919), Napoli et al. (PMID: 18977467), Ciolli et al. (PMID: 33480224)
Thanks for your specific literature suggestions. We have included the Stramazzo paper. As to the Napoli paper, we found a study about gene polymorphism, estrogen metabolism and bone density. We could not establish anything supporting our paper, so we did not cite it. As to the paper by Ciolli, we found a study about navigated percutaneous screw fixation in pelvic fractures. We also could not establish anything supporting our paper in this very interesting study, so we also did not cite it. The literature was updated thoroughly. Thanks for your very helpful comments.
Methods
Well written. Specify what type surgery lower limb.
Thanks for letting us know. We have pointed it out more clearly.
Discussion
well written
Thank you very much. From an expert, that is a very meaningful comment!
Round 2
Reviewer 1 Report
The manuscript has provided a good response to the questions raised by the reviewers. This is a very interesting study, well written, relevant, and fluent in line with the requirements of the journal. I think it is acceptable.